# Trading Complexity for Sparsity
# in Random Forest Explanations

## Abstract

Random forests have long been considered as powerful model ensembles in machine learning. By training multiple decision trees, whose diversity is fostered through data and feature subsampling, the resulting random forest can lead to more stable and reliable predictions than a single decision tree. This however comes at the cost of decreased interpretability: while decision trees are often easily interpretable, the predictions made by random forests are much more difficult to understand, as they involve a majority vote over hundreds of decision trees. In this paper, we examine different types of *reasons* that explain "why" an input instance is classified as positive or negative by a Boolean random forest. Notably, as an alternative to *sufficient reasons* taking the form of prime implicants of the random forest, we introduce *majoritary reasons* which are prime implicants of a strict majority of decision trees. For these different abductive explanations, the tractability of the generation problem (finding one reason) and the minimization problem (finding one shortest reason) are investigated. Experiments conducted on various datasets reveal the existence of a trade-off between runtime complexity and sparsity. Sufficient reasons - for which the identification problem is $\mathsf{DP}$-complete - are slightly larger than majoritary reasons that can be generated using a simple linear-time greedy algorithm, and significantly larger than *minimal* majoritary reasons that can be approached using an anytime PARTIAL MAXSAT algorithm.

## 1 Introduction

Over the past two decades, rapid progress in statistical machine learning has led to the deployment of models endowed with remarkable predictive capabilities. Yet, as the spectrum of applications using statistical learning models becomes increasingly large, explanations for why a model is making certain predictions are ever more critical. For example, in medical diagnosis, if some model predicts that an image is malignant, then the doctor may need to know which features in the image have led to this classification. Similarly, in the banking sector, if some model predicts that a customer is a fraud, then the banker might want to know why. Therefore, having explanations for why certain predictions are made is essential for securing user confidence in machine learning technologies [21, 22].

This paper focuses on classifications made by *random forests*, a popular ensemble learning method that constructs multiple randomized decision trees during the training phase, and predicts by taking a majority vote over the base classifiers [8]. Since decision tree randomization is achieved by essentially coupling data subsampling (or bagging) and feature subsampling, random forests are fast and easy to implement, with few tuning parameters. Furthermore, they often make accurate and robust predictions in practice, even for small data samples and high-dimensional feature spaces [6]. For these reasons, random forests have been used in various applications including, among others, computer vision [11], crime prediction [7], ecology [12], genomics [9], and medical diagnosis [3].

Despite their success, random forests are much less interpretable than decision trees. Indeed, the prediction made by a decision tree on a given data instance can be easily interpreted by reading the unique root-to-leaf path that covers the instance. Contrastingly, there is no such *direct reason* in a random forest, since the prediction is derived from a majority vote over multiple decision trees. So, a key issue in random forests is to infer *abductive explanations*, that is, to explain in concise terms why a data instance is classified as positive or negative by the model ensemble.

**Related Work.** Explaining random forest predictions has received increasing attention in recent years [5, 10, 18]. Notably, in the classification setting, [10, 18] have focused on *sufficient reasons*, which are abductive explanations involving only relevant features [13]. More specifically, if we view any random forest classifier as a Boolean function $f$, then a sufficient reason for classifying a data instance $x$ as positive by $f$ is a *prime implicant* $t$ of $f$ covering $x$. By construction, removing any feature from a sufficient reason $t$ would question the fact that $t$ explains the way $x$ is classified by $f$. Interestingly, if $f$ is described by a single decision tree, then generating a sufficient reason for any input instance $x$ can be done in linear time. Yet, in the general case where $f$ is represented by an arbitrary number of decision trees, the problem of identifying a sufficient reason is DP-complete. Despite this intractability statement, the empirical results reported in [18] show that a MUS-based algorithm for computing sufficient reasons proves quite efficient in practice.

In addition to "model-based" explanations investigated in [10, 18], "model-agnostic" explanations can be applied to random forests. Notably, the LIME method [27] extrapolates a linear threshold function $g$ from the behavior of the random forest $f$ around an input instance $x$. Yet, even if a prime implicant of the linear threshold function can be easily computed, this explanation is *not* guaranteed abductive since $g$ is only an approximation of $f$.

**Contributions.** In this paper, we introduce several new notions of abductive explanations: *direct reasons* extend to the case of random forests the corresponding notion defined primarily for decision trees, and *majority reasons* are weak forms of abductive explanations which take into account the averaging rule of random forests. Informally, a majority reason for classifying a instance $x$ as positive by some random forest $f$ is a prime implicant $t$ of a majority of decision trees in $f$ that covers $x$. Thus, any sufficient reason is a majority reason, but the converse is not true. For these different reasons, we examine the tractability of both the generation (finding one explanation) and the minimization (finding one shortest explanation) problems. To the best of our knowledge, all complexity results related to random forest explanations are new, if we make an exception for the intractability of generating sufficient reasons, which was recently established in [18]. Notably, direct reasons and majority reasons can be derived in time polynomial in the size of the input (the instance and the random forest used to classify it). By contrast, the identification of minimal majority reasons is NP-complete, and the identification of minimal sufficient reasons is $\Sigma_2^p$-complete.

Based on these results, we provide algorithms for deriving random forest explanations, which open the way for an empirical comparison. Our experiments made on standard benchmarks show the existence of a trade-off between the runtime complexity of finding (possibly minimal) abductive explanations and the sparsity of such explanations. In a nutshell, majority reasons and minimal majority reasons offer interesting compromises in comparison to, respectively, sufficient reasons and minimal sufficient reasons. Indeed, the size of majority reasons and the computational effort required to generate them are generally smaller than those obtained for sufficient reasons. Furthermore, minimal majority reasons outperform minimal sufficient reasons, since the latter are too computationally demanding. In fact, using an *anytime* PARTIAL MAXSAT solver for minimizing majority reasons, we derive sparse explanations which are typically much shorter than all other forms of abductive explanations. Proofs and additional empirical results are provided as supplementary material.

## 2 Preliminaries

For an integer $n$, let $[n] = \{1, \cdots, n\}$. By $\mathcal{F}_n$ we denote the class of all Boolean functions from $\{0, 1\}^n$ to $\{0, 1\}$, and we use $X_n = \{x_1, \cdots, x_n\}$ to denote the set of input Boolean variables. Any Boolean vector $x \in \{0, 1\}^n$ is called an *instance*. For any function $f \in \mathcal{F}_n$, an instance $x \in \{0, 1\}^n$ is called a *positive example* of $f$ if $f(x) = 1$, and a *negative example* otherwise.

We refer to $f$ as a propositional formula when it is described using the Boolean connectives $\wedge$ (conjunction), $\vee$ (disjunction) and $\neg$ (negation), together with the constants 1 (true) and 0 (false). As

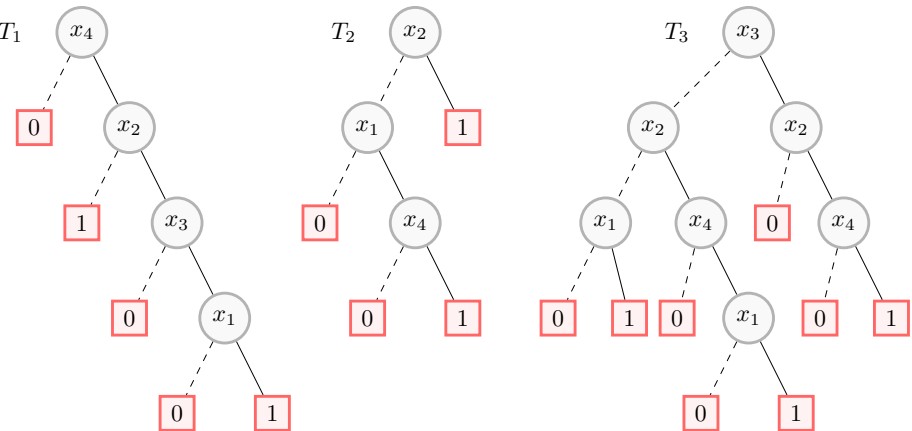

Figure 1: A random forest $F = \{T_1, T_2, T_3\}$ for recognizing *Cattleya* orchids. The left (resp. right) child of any decision node labelled by $x_i$ corresponds to the assignment of $x_i$ to 0 (resp. 1).

usual, a *literal* $l_i$ is a variable $x_i$ or its negation $\neg x_i$, also denoted $\overline{x}_i$. A *term* (or *monomial*) $t$ is a conjunction of literals, and a *clause* $c$ is a disjunction of literals. A DNF *formula* is a disjunction of terms and a CNF *formula* is a conjunction of clauses. The set of variables occurring in a formula $f$ is denoted $Var(f)$. In the rest of the paper, we shall often treat instances as terms, and terms as sets of literals. Given an assignment $\boldsymbol{z} \in \{0,1\}^n$, the corresponding term is defined as

$$t_{\boldsymbol{z}} = \bigwedge_{i=1}^{n} x_i^{z_i} \text{ where } x_i^0 = \overline{x}_i \text{ and } x_i^1 = x_i$$

A term $t$ *covers* an assignment $\boldsymbol{z}$ if $t \subseteq t_{\boldsymbol{z}}$. An *implicant* of a Boolean function $f$ is a term that implies $f$, that is, a term $t$ such that $f(\boldsymbol{z}) = 1$ for every assignment $\boldsymbol{z}$ covered by $t$. A *prime implicant* of $f$ is an implicant $t$ of $f$ such that no proper subset of $t$ is an implicant of $f$.

With these basic notions in hand, a (Boolean) *decision tree* on $X_n$ is a binary tree $T$, each of whose internal nodes is labeled with one of $n$ input variables, and whose leaves are labeled 0 or 1. Every variable is supposed (w.l.o.g.) to occur at most once on any root-to-leaf path (read-once property). The value $T(\boldsymbol{x}) \in \{0,1\}$ of $T$ on an input instance $\boldsymbol{x}$ is given by the label of the leaf reached from the root as follows: at each node go to the left or right child depending on whether the input value of the corresponding variable is 0 or 1, respectively. A (Boolean) *random forest* on $X_n$ is an ensemble $F = \{T_1, \cdots, T_m\}$, where each $T_i$ ($i \in [m]$) is a decision tree on $X_n$, and such that the value $F(\boldsymbol{x}) \in \{0,1\}$ on an input instance $\boldsymbol{x}$ is given by

$$F(\boldsymbol{x}) = \begin{cases} 1 & \text{if } \frac{1}{m}\sum_{i=1}^{m} T_i(\boldsymbol{x}) > \frac{1}{2} \\ 0 & \text{otherwise.} \end{cases}$$

The size of $F$ is given by $|F| = \sum_{i=1}^{m} |T_i|$, where $|T_i|$ is the number of nodes occurring in $T_i$. The class of decision trees on $X_n$ is denoted $\mathtt{DT}_n$, and the class of random forests with at most $m$ decision trees (with $m \geq 1$) over $\mathtt{DT}_n$ is denoted $\mathtt{RF}_{n,m}$. $\mathtt{RF}_n$ is the union of all $\mathtt{RF}_{n,m}$ for $m \in \mathbb{N}$.

**Example 1.** *The random forest $F = \{T_1, T_2, T_3\}$ in Figure 1 is composed of three decision trees. It separates Cattleya orchids from other orchids using the following features: $x_1$: "has fragrant flowers", $x_2$: "has one or two leaves", $x_3$: "has large flowers", and $x_4$: "is sympodial".*

It is well-known that any decision tree $T$ can be transformed into its negation $\neg T \in \mathtt{DT}_n$, by simply reverting the label of leaves. Negating a random forest can also be achieved in polynomial time:

**Proposition 1.** *There exists a linear-time algorithm that computes a random forest $\neg F \in \mathtt{RF}_{n,m}$ equivalent to the negation of a given random forest $F \in \mathtt{RF}_{n,m}$.*

Another important property of decision trees is that any $T \in \mathtt{DT}_n$ can be transformed in linear time into an equivalent disjunction of terms $\mathtt{DNF}(T)$, where each term coincides with a 1-path (i.e., a path from the root to a leaf labeled with 1), or a conjunction of clauses $\mathtt{CNF}(T)$, where each clause is the negation of term describing a 0-path. When switching to random forests, the picture is quite different:

**Proposition 2.** *Any CNF or DNF formula can be converted in linear time into an equivalent random forest, but there is no polynomial-space translation from RF to CNF or to DNF.*

## 3 Random Forest Explanations

The key focus of this study is to explain *why* a given (Boolean) random forest classifies some incoming data instance as positive or negative. This calls for a notion of abductive explanation[1]. Formally, given a Boolean function $f \in \mathcal{F}_n$ and an instance $\boldsymbol{x} \in \{0,1\}^n$, an *abductive explanation* for $\boldsymbol{x}$ given $f$ is an implicant $t$ of $f$ (resp. $\neg f$) if $f(\boldsymbol{x}) = 1$ (resp. $f(\boldsymbol{x}) = 0$) that covers $\boldsymbol{x}$. An abductive explanation $t$ for $\boldsymbol{x}$ given $f$ always exists, since $t = t_{\boldsymbol{x}}$ is such a (trivial) explanation. So, in the rest of this section, we shall mainly concentrate on *sparse* forms of abductive explanations.

Before delving into details, it is worth mentioning that if $f$ is represented by a random forest then, without loss of generality, we can focus on the case where $\boldsymbol{x}$ is a positive example of $f$, because $\neg f$ can be computed in linear time (by Proposition 1). Nevertheless, for the sake of clarity, we shall consider both cases $f(\boldsymbol{x}) = 1$ and $f(\boldsymbol{x}) = 0$ in our definitions.

### 3.1 Direct Reasons

For a decision tree $T \in \mathtt{DT}_n$ and a data instance $\boldsymbol{x} \in \{0,1\}^n$, the *direct reason* of $\boldsymbol{x}$ given $T$ is the term $t_{\boldsymbol{x}}^T$ corresponding to the unique root-to-leaf path of $T$ that covers $\boldsymbol{x}$. We can extend this simple form of abductive explanation to random forests as follows:

**Definition 1.** *Let $F = \{T_1, \ldots, T_m\}$ be a random forest in $\mathtt{RF}_{n,m}$, and $\boldsymbol{x} \in \{0,1\}^n$ be an instance. Then, the* direct reason *for $\boldsymbol{x}$ given $F$ is the term $t_{\boldsymbol{x}}^F$ defined by*

$$t_{\boldsymbol{x}}^F = \begin{cases} \bigwedge_{T_i \in F : T_i(\boldsymbol{x}) = 1} t_{\boldsymbol{x}}^{T_i} & \text{if } F(\boldsymbol{x}) = 1 \\ \bigwedge_{T_i \in F : T_i(\boldsymbol{x}) = 0} t_{\boldsymbol{x}}^{T_i} & \text{if } F(\boldsymbol{x}) = 0 \end{cases}$$

By construction, $t_{\boldsymbol{x}}^F$ is an abductive explanation which can be computed in $\mathcal{O}(|F|)$ time.

**Example 2.** *Considering Example 1 again, the instance $\boldsymbol{x} = (1,1,1,1)$ is recognized as a Cattleya orchid, since $F(\boldsymbol{x}) = 1$. The direct reason for $\boldsymbol{x}$ given $F$ is $t_{\boldsymbol{x}}^F = x_1 \wedge x_2 \wedge x_3 \wedge x_4$. It coincides with $t_{\boldsymbol{x}}$. Consider now the instance $\boldsymbol{x}' = (0,1,0,0)$; it is not recognized as a Cattleya orchid, since $F(\boldsymbol{x}) = 0$. The direct reason for $\boldsymbol{x}'$ given $F$ is $t_{\boldsymbol{x}'}^F = x_2 \wedge \overline{x}_3 \wedge \overline{x}_4$. It is a better abductive explanation than $t_{\boldsymbol{x}'}$ itself since it does not contain $\overline{x}_1$, which is locally irrelevant.*

### 3.2 Sufficient Reasons

Another valuable notion of abductive explanation is the one of *sufficient reason*[2], defined for any Boolean classifier [13]. In the setting of random forests, such explanations can be defined as follows:

**Definition 2.** *Let $F \in \mathtt{RF}_n$ be a random forest and $\boldsymbol{x} \in \{0,1\}^n$ be an instance. A* sufficient reason *for $\boldsymbol{x}$ given $F$ is a prime implicant $t$ of $F$ (resp. $\neg F$) if $F(\boldsymbol{x}) = 1$ (resp. $F(\boldsymbol{x}) = 0$) that covers $\boldsymbol{x}$.*

**Example 3.** *For our running example, $x_1 \wedge x_2 \wedge x_4$ and $x_3 \wedge x_4$ are the sufficient reasons for $\boldsymbol{x}$ given $F$. $\overline{x}_4$ and $\overline{x}_1 \wedge x_2 \wedge \overline{x}_3$ are the sufficient reasons for $\boldsymbol{x}'$ given $F$.*

Unlike arbitrary abductive explanations, all features occurring in a sufficient reason $t$ are *relevant*. Indeed, removing any literal from $t$ would question the fact that $t$ implies $F$. To this point, the direct reason $t_{\boldsymbol{x}}^F$ for $\boldsymbol{x}$ given $F$ may contain arbitrarily many more features than a sufficient reason for $\boldsymbol{x}$ given $F$, since this was already shown in the case where $F$ consists in a single decision tree [17].

The problem of finding a sufficient reason $t$ for an input instance $\boldsymbol{x} \in \{0,1\}^n$ with respect to a given random forest $F \in \mathtt{RF}_n$, has recently been shown $\mathsf{DP}$-complete [18]. In fact, even the apparently simple task of *checking* whether $t$ is an implicant of $F$ is already hard:

**Proposition 3.** *Let $F$ be a random forest in $\mathtt{RF}_n$ and $t$ be a term over $X_n$. Then, deciding whether $t$ is an implicant of $F$ is $\mathsf{coNP}$-complete.*

The above result is in stark contrast with the computational complexity of checking whether a term $t$ is an implicant of a decision tree $T$. This task can be solved in polynomial time, using the fact that

---

[1]Unlike [15], we do not require those explanations to be minimal w.r.t. set inclusion, in order to keep the concept distinct (and actually more general) then the one of sufficient reasons.

[2]Sufficient reasons are also known as prime-implicant explanations [29].

$T$ can be converted (in linear time) into its clausal form $\mathrm{CNF}(T)$, together with the fact that testing whether $t$ implies $\mathrm{CNF}(T)$ can be done in $\mathcal{O}(|T|)$ time. That mentioned, in the case of random forests, the implicant test can be achieved via a call to a SAT oracle:

**Proposition 4.** *Let $F = \{T_1, \ldots, T_m\}$ be a random forest of $\mathrm{RF}_{n,m}$, and $t$ be a (satisfiable) term over $X_n$. Let $H$ be the $\mathrm{CNF}$ formula*

$$\{(\overline{y}_i \vee c) : i \in [m], c \in \mathrm{CNF}(\neg T_i)\} \cup \mathrm{CNF}\left(\sum_{i=1}^{m} y_i > \frac{m}{2}\right)$$

*where $\{y_1, \ldots, y_m\}$ are fresh variables and $\mathrm{CNF}\left(\sum_{i=1}^{m} y_i > \frac{m}{2}\right)$ is a $\mathrm{CNF}$ encoding of the cardinality contraint $\sum_{i=1}^{m} y_i > \frac{m}{2}$. Then, $t$ is an implicant of $F$ if and only if $H \wedge t$ is unsatisfiable.*

Based on such an encoding, the sufficient reasons for an instance $\boldsymbol{x}$ given a random forest $F$ can be characterized in terms of MUS (minimal unsatisfiable subsets), as suggested in [18]. This characterization is useful because many SAT-based algorithms for computing a MUS (or even all MUSes) of a $\mathrm{CNF}$ formula have been pointed out for the past decade [2, 19, 20], and hence, one can take advantage of them for computing sufficient reasons.

Going one step further, a natural way for improving the clarity of sufficient reasons is to focus on those of minimal size. Specifically, given $F \in \mathrm{RF}_n$ and $\boldsymbol{x} \in \{0, 1\}^n$, a *minimal sufficient reason* for $\boldsymbol{x}$ with respect to $F$ is a sufficient reason for $\boldsymbol{x}$ given $F$ of minimal size.[3]

**Example 4.** *For our running example, $x_3 \wedge x_4$ is the unique minimal sufficient reason for $\boldsymbol{x}$ given $F$, and $\overline{x}_4$ is the unique minimal reason for $\boldsymbol{x}'$ given $F$.*

As a by-product of the characterization of a sufficient reason in terms of MUS [18], a minimal sufficient reason for $\boldsymbol{x}$ given $f$ can be viewed as a *minimal* MUS. Thus, we can exploit algorithms for computing minimal MUSes (see e.g., [16]) in order to derive minimal sufficient reasons. However, deriving a minimal sufficient reason is computationally harder than deriving a sufficient reason:

**Proposition 5.** *Let $F \in \mathrm{RF}_n$, $\boldsymbol{x} \in \{0, 1\}^n$, and $k \in \mathbb{N}$. Then, deciding whether there exists a minimal sufficient reason $t$ for $\boldsymbol{x}$ given $F$ containing at most $k$ features is $\Sigma_2^p$-complete.*

### 3.3 Majoritary Reasons

Based on the above considerations, a natural question arises: does there exist a middle ground between direct reasons, which main contain many irrelevant features but are easy to calculate, and sufficient reasons, which only contain relevant features but are potentially much harder to generate? Inspired by the way prime implicants can be computed when dealing with decision trees, we can reply in the affirmative using the notion of *majoritary reasons*, defined as follows.

**Definition 3.** *Let $F = \{T_1, \ldots, T_m\}$ be a random forest in $\mathrm{RF}_{n,m}$ and $\boldsymbol{x} \in \{0, 1\}^n$ be an instance. Then, a* majoritary reason *for $\boldsymbol{x}$ given $F$ is a term $t$ covering $\boldsymbol{x}$, such that $t$ is an implicant of at least $\lfloor \frac{m}{2} \rfloor + 1$ decision trees $T_i$ (resp. $\neg T_i$) if $F(\boldsymbol{x}) = 1$ (resp. $F(\boldsymbol{x}) = 0$), and for every $l \in t$, $t \setminus \{l\}$ does not satisfy this last condition.*

**Example 5.** *For our running example, $\boldsymbol{x}$ has three majoritary reasons given $F$: $x_1 \wedge x_2 \wedge x_4$, $x_1 \wedge x_3 \wedge x_4$, and $x_2 \wedge x_3 \wedge x_4$. Those reasons are better than $t_{\boldsymbol{x}}^F$ in the sense that they are shorter than this direct reason. Contrastingly, $\boldsymbol{x}'$ has four majoritary reasons given $F$: $\overline{x}_1 \wedge \overline{x}_4$, $x_2 \wedge \overline{x}_4$, $\overline{x}_3 \wedge \overline{x}_4$, and $\overline{x}_1 \wedge x_2 \wedge \overline{x}_3$. Each of the two majoritary reasons $x_2 \wedge \overline{x}_4$, $\overline{x}_3 \wedge \overline{x}_4$ show that $t_{\boldsymbol{x}'}^F$ contains some irrelevant literals for the task of classifying $\boldsymbol{x}'$ using $F$.*

In general, the notions of majoritary reasons and of sufficient reasons do not coincide. Indeed, a sufficient reason $t$ is a prime implicant (covering $\boldsymbol{x}$) of the forest $F$, while a majoritary reason $t'$ is an implicant (covering $\boldsymbol{x}$) of a strict majority of decision trees in the forest $F$ satisfying the additional condition that $t'$ is a prime implicant of at least one of these decision trees. Viewing majoritary reasons as "weak" forms of sufficient reasons, they can include irrelevant features:

**Proposition 6.** *Let $F = \{T_1, \ldots, T_m\}$ be a random forest of $\mathrm{RF}_{n,m}$ and $\boldsymbol{x} \in \{0, 1\}^n$ such that $F(\boldsymbol{x}) = 1$. Unless $m < 3$, it can be the case that every majoritary reason for $\boldsymbol{x}$ given $F$ contains arbitrarily many more features than any sufficient reason for $\boldsymbol{x}$ given $F$.*

---

[3]Minimal sufficient reasons should not to be confused with *minimum-cardinality explanations* [29], where the minimality condition bears on the features set to 1 in the data instance $\boldsymbol{x}$.

209 What makes majority reasons valuable is that they are abductive and can be generated in linear time.
210 The evidence that any majority reason $t$ for $\boldsymbol{x}$ given $F$ is an abductive explanation for $\boldsymbol{x}$ given $F$
211 comes directly from the fact that if $t$ implies a majority of decision trees in $F$, then it is an implicant
212 of $F$ (note that the converse implication does not hold in general).

213 The tractability of generating majority reasons lies in the fact that they can be found using a simple
214 greedy algorithm. For the case where $F(\boldsymbol{x}) = 1$, start with $t = t_{\boldsymbol{x}}$, and iterate over the literals $l$ of $t$
215 by checking whether $t$ deprived of $l$ is an implicant of at least $\lfloor \frac{m}{2} \rfloor + 1$ decision trees of $F$. If so,
216 remove $l$ from $t$ and proceed to the next literal. Once all literals in $t_{\boldsymbol{x}}$ have been examined, the final
217 term $t$ is by construction an implicant of a strict majority of decision trees in $F$, such that removing
218 any literal from it would lead to a term that is no longer an implicant of this majority. So, $t$ is by
219 construction a majority reason. The case where $F(\boldsymbol{x}) = 0$ is similar, by simply replacing each
220 $T_i$ with its negation in $F$. This greedy algorithm runs in $\mathcal{O}(n|F|)$ time, using the fact that, on each
221 iteration, checking whether $t$ is an implicant of $T_i$ (for each $i \in [m]$) can be done in $\mathcal{O}(|T_i|)$ time.

222 By analogy with minimal sufficient reasons, a natural way of improving the quality of majority
223 reasons is to seek for shortest ones. Let $F \in \mathtt{RF}_n$ be a random forest and $\boldsymbol{x} \in \{0,1\}^n$ be an instance.
224 Then, a *minimal majority reason* for $\boldsymbol{x}$ given $F$ is a minimal-size majority reason for $\boldsymbol{x}$ given $F$.

225 **Example 6.** *For our running example, the three majority reasons for $\boldsymbol{x}$ given $F$ are its minimal*
226 *majority reasons. Contrastingly, among the majority reasons for $\boldsymbol{x}'$ given $F$, only $\overline{x}_1 \wedge \overline{x}_4$,*
227 *$x_2 \wedge \overline{x}_4$, and $\overline{x}_3 \wedge \overline{x}_4$ are minimal majority reasons.*

228 Unsurprisingly, the optimization task for majority reasons is more demanding than the generation
229 task. Yet, minimal majority reasons are easier to find than minimal sufficient reasons. Specifically:

230 **Proposition 7.** *Let $F \in \mathtt{RF}_n$, $\boldsymbol{x} \in \{0,1\}^n$, and $k \in \mathbb{N}$. Then, deciding whether there exists a*
231 *minimal majority reason $t$ for $\boldsymbol{x}$ given $F$ containing at most $k$ features is* NP-*complete.*

232 A common approach for handling NP-optimization problems is to rely on modern constraint solvers.
233 From this perspective, recall that a PARTIAL MAXSAT problem consists of a pair $(C_{\text{soft}}, C_{\text{hard}})$
234 where $C_{\text{soft}}$ and $C_{\text{hard}}$ are (finite) sets of clauses. The goal is to find a Boolean assignment that
235 maximizes the number of clauses $c$ in $C_{\text{soft}}$ that are satisfied, while satisfying all clauses in $C_{\text{hard}}$.

236 **Proposition 8.** *Let $F \in \mathtt{RF}_{n,m}$ and $\boldsymbol{x} \in \{0,1\}^n$ be an instance such that $F(\boldsymbol{x}) = 1$. Let*
237 *$(C_{\text{soft}}, C_{\text{hard}})$ be an instance of the* PARTIAL MAXSAT *problem such that:*

$$C_{\text{soft}} = \{\overline{x}_i : x_i \in t_{\boldsymbol{x}}\} \cup \{x_i : \overline{x}_i \in t_{\boldsymbol{x}}\}$$

$$C_{\text{hard}} = \{(\overline{y}_i \vee c_{|\boldsymbol{x}}) : i \in [m], c \in \mathtt{CNF}(T_i)\} \cup \mathtt{CNF}\left(\sum_{i=1}^m y_i > \frac{m}{2}\right)$$

238 *where $c_{|\boldsymbol{x}} = c \cap t_{\boldsymbol{x}}$ is the restriction of $c$ to the literals in $t_{\boldsymbol{x}}$, $\{y_1, \ldots, y_m\}$ are fresh variables and*
239 *$\mathtt{CNF}(\sum_{i=1}^m y_i > \frac{m}{2})$ is a* CNF *encoding of the contraint $\sum_{i=1}^m y_i > \frac{m}{2}$. The intersection of $t_{\boldsymbol{x}}$ with*
240 *$t_{\boldsymbol{z}^*}$, where $\boldsymbol{z}^*$ is an optimal solution of $(C_{\text{soft}}, C_{\text{hard}})$, is a minimal majority reason for $\boldsymbol{x}$ given $F$.*

241 Clearly, in the case where $F(\boldsymbol{x}) = 0$, it is enough to consider the same instance of PARTIAL MAXSAT
242 as above, except that $C_{\text{hard}} = \{(\overline{y}_i \vee c_{|\boldsymbol{x}}) : i \in [m], c \in \mathtt{CNF}(\neg T_i)\} \cup \mathtt{CNF}(\sum_{i=1}^m y_i > \frac{m}{2})$.

243 Thanks to this characterization result, one can leverage the numerous algorithms that have been
244 developed so far for PARTIAL MAXSAT (see e.g. [1, 23, 24, 28]) in order to compute minimal
245 majority reasons. We took advantage of it to achieve some of the experiments reported in Section 4.

# 4 Experiments

247 **Empirical setting.** The empirical protocol was as follows. We have considered 15 datasets, which
248 are standard benchmarks from the well-known repositories Kaggle (www.kaggle.com), OpenML
249 (www.openml.org), and UCI (archive.ics.uci.edu/ml/). These datasets are *compas*, *placement*,
250 *recidivism*, *adult*, *ad_data*, *mnist38*, *mnist49*, *gisette*, *dexter*, *dorothea*, *farm-ads*, *higgs_boson*,
251 *christine*, *gina*, and *bank*. *mnist38* and *mnist49* are subsets of the *mnist* dataset, restricted to the
252 instances of 3 and 8 (resp. 4 and 9) digits. Due to space constraints, additional information about
253 the datasets (especially the numbers and types of features, the number of instances), and about the
254 random forests that have been trained (especially, the number of Boolean features used, the number

of trees, the depth of the trees, the mean accuracy) are reported as a supplementary material. We used only datasets for binary classification, which is a very common kind of dataset. Categorical features have been treated as arbitrary numbers (the scale is nominal). As to numeric features, no data preprocessing has taken place: these features have been binarized on-the-fly by the random forest learning algorithm that has been used.

For every benchmark $b$, a 10-fold cross validation process has been achieved. Namely, a set of 10 random forest $F_b$ have been computed and evaluated from the labelled instances of $b$, partitioned into 10 parts. One part was used as the test set and the remaining 9 parts as the training set for generating a random forest. The classification performance for $b$ was measured as the mean accuracy obtained over the 10 random forests generated from $b$. As to the random forest learner, we have used the implementation provided by the Scikit-Learn [26] library in his version 0.23.2. The maximal depth of any decision tree in a forest has been bounded at 8. All other hyper-parameters of the learning algorithm have been set to their default value except the number of trees. We made some preliminary tests for tuning this parameter in order to ensure that the accuracy is good enough. For each benchmark $b$, each random forest $F$, and a subset of 25 instances $x$ picked up at random in the corresponding test set (leading to 250 instances per dataset) we have run the algorithms described in Section 3 for deriving the direct reason for $x$ given $F$, a sufficient reason for $x$ given $F$, a majority reason $x$ given $F$, a minimal majority reason for $x$ given $F$, and a minimal sufficient reason for $x$ given $F$.

For computing sufficient reasons and minimal majority reasons, we took advantage of the Pysat library [14] (version 0.1.6.dev15) which provides the implementation of the RC2 PARTIAL MAXSAT solver and an interface to MUSER [4]. When deriving majority reasons, we picked up uniformly at random 50 permutations of the literals describing the instance and tried to eliminate those literals (within the greedy algorithm) following the ordering corresponding to the permutation. As a majoritary reason for the instance, we kept a smallest reason among those that have been derived (of course, the corresponding computation time that has been measured is the cumulated time over the 50 tries). Sufficient reasons have been computed as MUSes, as explained before.

We also derived a "LIME explanation" for each instance. Such an explanation has been generated thanks to the following approach. For any $x$ under consideration, one first used LIME [27] to generate an associated linear model $w_x$ where $w_x \in \mathbb{R}^n$. This linear model $w_x$ classifies any instance $x'$ as a positive instance if and only if $w_x \cdot x' > 0$. Furthermore, $w_x$ classifies the instance to be explained $x$ in the same way as the black box model considered at start (in our case, the random forest $F$). We ran the LIME implementation linked to [27] in its latest version. Interestingly, a minimal sufficient reason $t$ for $x$ given $w_x$ can be generated in polynomial time from $w_x$. We call it a LIME explanation for $x$. The computation of $t$ is as follows. If $x$ is classified positively by $w_x$, in order to derive $t$, it is enough to sum in a decreasing way the positive weights $w_i$ occurring in $w_x$ until this sum exceeds the sum of the opposites of all the negative weights occurring in $w_x$. The term $t$ composed of the variables $x_i$ corresponding to the positive weights that have been selected is by construction a minimal sufficient reason for $x$ given $w_x$ since for every $x'$ covered by $t$, the inequation $w_x \cdot x' > 0$ necessarily holds; indeed, it holds in the worst situation where all the variables associated with a positive weight in $w_x$ and not belonging to $t$ are set to 0, whilst all the variables associated with a negative weight in $w_x$ are set to 1. Similarly, if $x$ is classified negatively by $w_x$, in order to derive $t$, it is enough to sum in an increasing way the negative weights $w_i$ occurring in $w_x$ until this sum is lower than or equal to the opposite of the sum of all the positive weights occurring in $w_x$. This time, the term $t$ composed of the variables $x_i$ corresponding to the negative weights that have been selected is by construction a minimal sufficient reason for $x$ given $w_x$.

All the experiments have been conducted on a computer equipped with Intel(R) XEON E5-2637 CPU @ 3.5 GHz and 128 Gib of memory. A time-out (TO) of 600s has been considered for each instance and each type of explanation, except LIME explanations.

**Results.** A first conclusion that can be drawn from our experiments is the intractability of computing in practice minimal sufficient reasons (this is not surprising, since this coheres with the complexity result given by Proposition 5). Indeed, we have been able to compute within the time limit of 600s a minimal reason for only 10 instances and a single dataset (*compas*).

Due to space limitations, we report hereafter empirical results about two datasets only, namely *placement* and *gisette* (the results obtained on the other datasets are similar and available as a

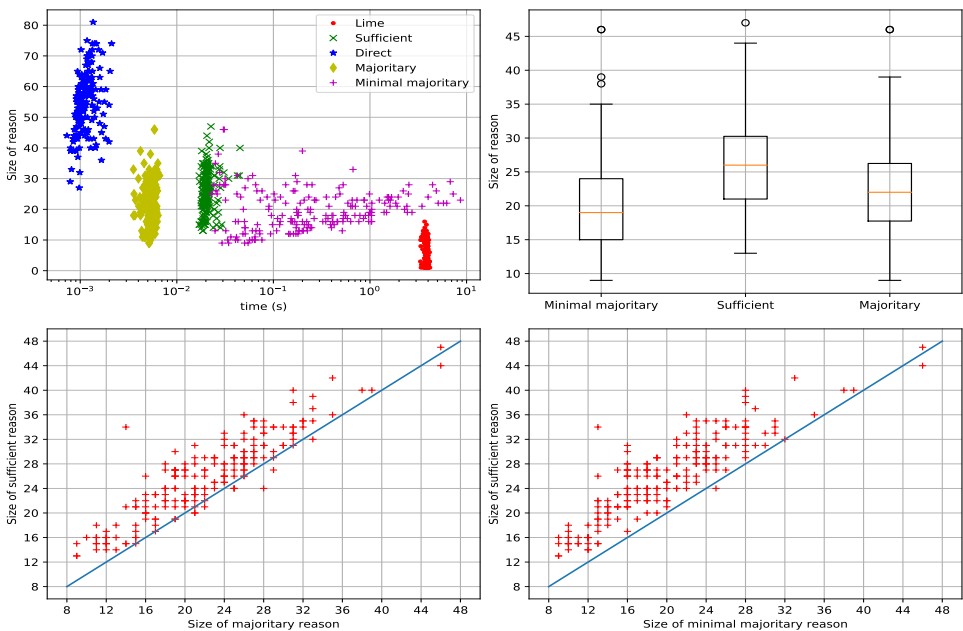

Figure 2: Empirical results for the *placement* dataset.

supplementary material). The *placement* data set is about the placement of students in a campus. It consists of 215 labelled instances. Students are described using 13 features, related to their curricula, the type and work experience and the salary. An instance is labelled as positive when the student gets a job. The random forest that has been generated consists of 25 trees, and its mean accuracy is 97.6%. *gisette* is a much larger dataset, based on 5000 features and containing 7000 labelled instances. Features correspond to pixels. The problem is to separate the highly confusible digits 4 and 9. An instance is labelled as positive whenever the picture represents a 9. The random forest that has been generated consists of 85 trees, and its mean accuracy is 96%.

Figure 2 provides the results obtained for *placement*, using four plots. Each dot represents an instance. The first plot shows the time needed to compute a reason on the x-axis, and the size of this reason on the y-axis. On this plot, no dot corresponds to a minimal sufficient reason because their computation did not terminate before the time-out. The plot also highlights that all the other reasons have been computed within the time limit, and in general using a small amount of time. In particular, it shows that the direct reason can be quite large, that the computation of LIME explanations is usually more expensive than the ones of the other explanations, and that LIME explanations can be very short (but one must keep in mind that they are not abductive explanations in general[4]). A box plot about the sizes of all the explanations is reported (the LIME ones and the direct reasons are not presented for the sake of readibility). The figure also provides two scatter plots, aiming to compare the size of majority reasons with the size of sufficient reasons, as well as the size of the minimal majority reasons with the size of sufficient reasons. These plots clearly show the benefits that can be offered by considering majority reasons and minimal majority reasons instead of sufficient reasons.

Figure 3 synthesizes the results obtained for *gisette*, using four plots again. Three of them are of the same kind as the plots used for *placement*. Conclusions similar to those drawn for *placement* can be derived for *gisette*, with some exceptions. First of all, this time, no dot corresponds to a minimal majority reason because their computation did not terminate before the time-out. Furthermore, LIME explanations are very long here. This can be explained by the fact that the computation achieved by LIME relies on a binary representation of the instance that is quite different (and possibly much larger) than the one considered in the representation of the random forest. Indeed, each decision tree of the forest focuses only on a subset of most important features (in the sense of Gini criterion) found during the learning phase. In our experiments, the size of LIME explanations was typically high for datasets based on many features.

---

[4]See also [25] that reports some experiments about ANCHOR (the successor of LIME), assessing the quality of the explanations computed using ANCHOR.

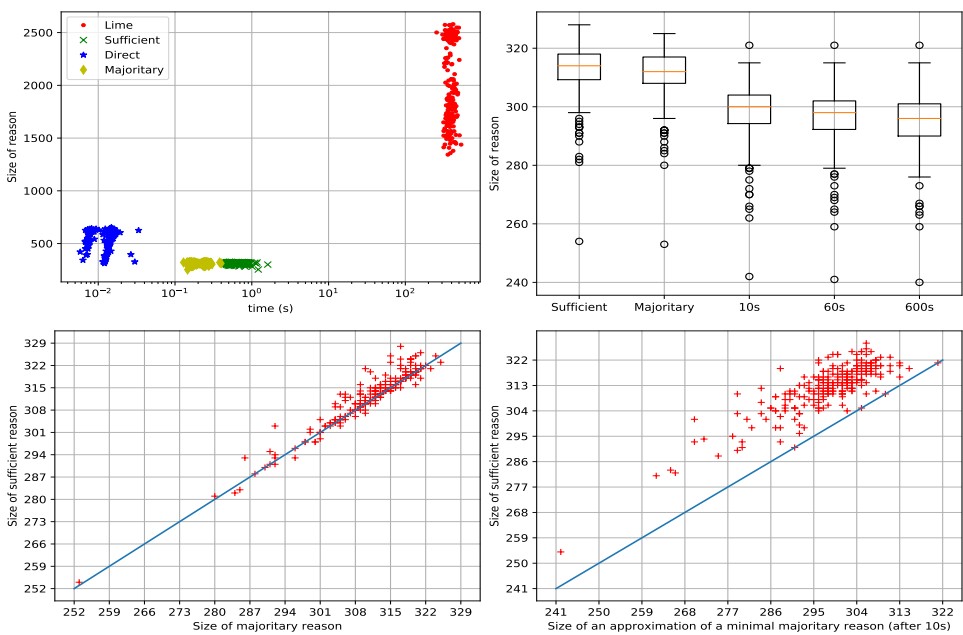

Figure 3: Empirical results for the *gisette* dataset.

When minimal majority reasons are hard to be computed (as it is the case for *gisette*), an approach consists in approximating them. Interestingly, one can take advantage of an incremental PARTIAL MAXSAT ALGORITHM, like LMHS [28], to do the job. Specifically, the result given in Proposition 8 provides a way to derive abductive explanations for an instance $x$ given a random forest $F$ in an *anytime* fashion. Basically, using LMHS, a Boolean assignment $z$ satisfying all the hard constraints of $C_{\mathrm{hard}}$ and a given number, say $k$, of soft constraints from $C_{\mathrm{soft}}$ is looked for ($k$ is set to 0 at start). If such an assignment is found, then one looks for an assignment satisfying $k + 1$ soft constraint, and so on, until an optimal solution is found or a preset time bound is reached. In many cases, the most demanding step from a computational standpoint is the one for which $k$ is the optimal value (but one ignores it) and one looks for an assignment that satisfies $k + 1$ soft constraint (and such an assignment does not exist). By construction, every $z$ that is generated that way is such that $t_{x} \cap t_{z}$ is an implicant of $F$ that covers $x$ (and hence, an abductive explanation). The approximation $z$ of a minimal majority reason for $x$ given $F$, which is obtained when the time limit is met, can be significantly shorter than the sufficient reason for $x$ given $F$ that has been derived. In our experiments, we used three time limits: 10s, 60s, 600s. As the box plot and the dedicated scatter plot given in Figure 3 show it, the sizes of the approximations $z$ which are derived gently decrease with time. Interestingly, the size savings that are achieved in comparison to sufficient reasons are significant, even for the smallest time bound of 10s that has been considered.

## 5   Conclusion

In this paper, we have introduced, analyzed and evaluated some new notions of abductive explanations suited to random forest classifiers, namely majority reasons and minimal majority reasons. Our investigation reveals the existence of a trade-off between runtime complexity and sparsity for abductive explanations. Unlike sufficient reasons, majority reasons and minimal majority reasons may contain irrelevant features. Despite this evidence, majority reasons and minimal majority reasons appear as valuable alternative to sufficient reasons. Indeed, majority reasons can be computed in polynomial time while sufficient reasons cannot (unless $\mathsf{P} = \mathsf{NP}$). In addition, most of the time in our experiments, majority reasons appear as slightly smaller than sufficient reasons. Minimal majority reasons can be looked for when majority reasons are too large, but this is at the cost of an extra computation time that can be important, and even prohibitive in some cases. However, minimal majority reasons can be approximated using an *anytime* PARTIAL MAXSAT algorithm. Empirically, approximations can be derived within a small amount of time and their sizes are significantly smaller than the ones of sufficient reasons.

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
