# OpenReview forum: "Trading Complexity for Sparsity in Random Forest Explanations"
_NeurIPS.cc/2021/Conference — NeurIPS 2021 Submitted_

### Official Review · Reviewer_4ukE · 2021-07-11

**Rating:** 3
**Confidence:** 5

**Summary:**

This paper is devoted to computing abductive explanations for random
forest (RF) machine learning (ML) models. Concretely, the paper
proposes a variant of such explanations called majoritary reasons and
argues that this new kind of explanation is not only easier to compute
than computing sufficient reasons (both from the theoretical and
practical points of view) but also that in practice they may be
advantageous to a user due to their often smaller size than that of
sufficient reasons. For that, an encoding and a MaxSAT-based solution
for computing majoritary reasons for RF models is proposed.


**Limitations And Societal Impact:**

See the main comments above.

**Main Review:**

This work studies the issue of explainability, which is arguably one
of the most important problems in the current state of affairs in AI.
More importantly, this paper makes an attempt to approach this problem
from the right viewpoint - namely, from the perspective of logic and
formal reasoning, which is in clear contrast to all the "magic"
heuristic methods being studied in the vast majority of works on XAI.
Furthermore, random forests represent an important and quite widely
used ML model, which further underscores the results reported in the
paper.

Having said that, I feel somewhat negative about this paper as I have
a few issues with it. First, although the list of claimed
contributions is large, not all of them seem to be necessary in the
context of the paper and the conference. As an example, Propositions 1
and 2 look unrelated to the rest of the paper. Furthermore, I am not
sure to what extent Proposition 2 is true because one can trivially
encode an RF to a CNF formula in polynomial time and space using
auxiliary variables, and the authors themselves do mention this later
in the paper. Here, if the authors meant some concrete form of
reduction to CNF/DNF, they should have been clear.

I take Proposition 3 as a contribution even though it might be seen as
somewhat simple. Also, abductive and contrastive explanations are
known to be related with MUSes and MCSes of a logic representation of
the decision function, respectively. This makes Proposition 5 not
surprising as computing smallest size abductive explanations is in
general hard for the second level of the polynomial hierarchy, due to
the well-known complexity of SMUS. (Granted that the exact complexity
of this problem in the particular case of RF models was previously not
studied.)

The paper requires some proof-reading as some of the examples are
wrong. For example, {x_3=1, x_4=1} is not a sufficient reason for
prediction F(x)=1. One can get a simple counterexample by setting
x_1=0 and x_2=0, which makes T_2 and T_3 predict class 0 (hence, the
majority of trees report class 0, thus, making the RF predict 0).
This mistake invalidates some of the later examples, which refer to
this one.

My main issue with the paper is related to the main contribution,
namely, majoritary reasons. Concretely, I do not see why they are not
guaranteed to be sufficient reasons. Following the definition, such a
reason should imply that the majority of trees predict the right
class. By definition of RF, this means that such an explanation
suffices for the prediction of RF to hold, i.e. it is a sufficient
reason for RF. (Please let me know if I am missing something.) Hence,
I can't easily see why the following claims are correct, as well as
if/why the MaxSAT encoding and the use a MaxSAT solver is enough to
compute a majoritary reason. The "difference" between the two types of
explanations is mentioned on line 204 but is not discussed at all.
Also, the examples fail to properly illustrate the concept.

More on examples, there is not a simple example to illustrate the
encoding - and there must be such examples.

Also, I am not entirely convinced by the need to introduce the concept
of direct explanations - it is straightforward that one can make the
union of the paths firing the prediction in each tree predicting the
right class to derive a "common" reason for the prediction. Also,
abductive explanations are typically equated to sufficient reasons,
i.e. they are subset-minimal. Also, using the word "minimal" instead
of "minimum"/"smallest" is a bit confusing when the authors refer to
explanations of smallest size.

Regarding the experiments, I have a few comments:
	- what SAT solver do you use?
	- what configuration of a MaxSAT solver do you use?
	- how do you encode the cardinality constraints?
	- on line 287 you mention that sufficient reasons for a linear
	  classifier can be computed in polynomial time but fail to cite
	  the corresponding work.
	- there is no need to use LMHS (also, in what sense is it
	  incremental?) - you can simply use an effective incomplete
	  solver that over-approximates the exact solution well enough,
	  say Loandra.

Finally, I would like to note that I failed to find a supplementary
PDF file that would contain the proofs and/or extended discussion. The
zip-file contains the datasets, source code and other things related
to experimental results only. I look forward to the authors' rebuttal
and hope that the authors will be able to convincingly comment on the
issues I raised and/or prove me wrong.

**Time Spent Reviewing:**

10

---

> ### Author Response · Authors · 2021-08-10
> **Response to  the review by Reviewer 4ukE**
>
> Thanks for your comments.
>
> _’’Propositions 1 and 2 look unrelated to the rest of the paper.’’_
>
> Actually, this is not the case. Proposition 1 is used to prove Proposition 6, and Proposition 2 is used to prove Propositions 3 and 5.
>
> _’’Furthermore, I am not sure to what extent Proposition 2 is true because one can  trivially encode an RF to a CNF formula in polynomial time and space using auxiliary  variables.’’_
>
> We understand your point, but the use of auxiliary variables is forbidden; the resulting  CNF formula is expected to be logically equivalent to the input RF (and not only equivalent over the set of variables used in the RF). We confirm that Proposition 2  holds.
>
> _’’The paper requires some proof-reading as some of the examples are wrong. For  example, {x_3=1, x_4=1} is not a sufficient reason for prediction F(x)=1.’’_
>
> Apologies for this mistake. {$x_1=1, x_4=1$} is a sufficient reason for prediction $F(x)=1$.
>
> _’’My main issue with the paper is related to the main contribution, namely, majoritary  reasons. Concretely, I do not see why they are not guaranteed to be sufficient  reasons.’’_
>
> Sorry that it was not clear enough. The key observation is that even if every implicant of a formula $A$ (resp. $B$) is an implicant  of $A \lor B$, it is not (always) the case that every _prime_ implicant of $A$ (resp. $B$) is a _prime_ implicant of $A \lor B$. To this very point, consider our running example and take $\mathbf x =  (1,1,1,1)$. Here, $\mathbf t =$  {$x_1=1, x_3=1, x_4=1$} is a majoritary reason for $F(\mathbf x)=1$, since it covers  $\mathbf x$, it is a prime implicant of $T_1$ and an implicant of $T_2$ (note that it is not an implicant of $T_3$).  Thus, $\mathbf t$ is an implicant of $A = T_1 \land T_2$ (it is a prime one), and hence an implicant of $F$, where $F$ is equivalent to
> $$
> (T_1 \land T_2) \lor (T_1 \land T_3) \lor (T_2 \land T_3)
> $$
> However, $\mathbf t$ is _not_ a prime implicant of $F$. Indeed, the proper subset {$x_1=1, x_4=1$} is a sufficient reason for predicting $F(x)=1$ since {$x_1=1, x_4=1$} is a prime implicant of $F$ that covers $\mathbf x$. Why is it a prime implicant of $F$? Just because {$x_1=1, x_3=0, x_4=1$} is an implicant of $T_2 \land T_3$, hence an implicant of $B = (T_1 \land T_3) \lor (T_2  \land T_3)$ (more precisely,  it is a prime implicant of $B$), and so it is also an implicant of $F$.
>
> _’’More on examples, there is not a simple example to illustrate the encoding.’’_
>
> We will add such an example (we did not do it for space reasons).
>
> _’’Also, I am not entirely convinced by the need to introduce the concept of direct  explanations. Also, abductive explanations are typically equated to sufficient reasons, i.e. they are subset-minimal.’’_
>
> We agree that the concept of direct explanations is not a breakthrough. However, simplicity is not bad. From the empirical side, direct reasons may contain far less features than instances (see file data_RF in the supplementary material).
>
> In our paper, subset minimality is not required for abductive explanations (please see footnote 1). This relaxation of Def 3 in [16] coheres with early work about abduction in AI (see e.g.(Eiter & Gottlob, JACM, 1995)).
>
> _’’Regarding the experiments, I have a few comments:’’_
>
> _’’- what SAT solver do you use?’’_
>
> MUSes have been computed using muser2, that uses glucose3 as an underlying SAT solver. MaxSAT solutions have been computed using LMHS, which is based on minisat.
>
> _’’- what configuration of a MaxSAT solver do you use?’’_
>
> The default configuration.
>
> _’’- how do you encode the cardinality constraints?’’_
>
> Using a sequential counter (https://www.carstensinz.de/papers/CP-2005.pdf).
>
> _’’  - on line 287 you mention that sufficient reasons for a linear classifier can be  computed in polynomial time but fail to cite the corresponding work.’’_
>
> This is something new (as far as we know). So there is no reference to provide. Instead, from line 288 to line 300, we have presented an algorithm to derive such a (minimal) sufficient reason.
>
> _’’ - there is no need to use LMHS (also, in what sense is incremental?)’’_
>
> We have chosen LMHS as a MaxSAT solver because of its ''incremental behaviour’’, i.e., its capacity to generate suboptimal solutions in sequence, with an improvement at each step, before outputting an optimal solution (this gives LMHS an anytime flavor).
>
> _’’ - you can simply use an effective incomplete solver that over-approximates the exact  solution well enough, say Loandra.’’_
>
> Indeed, we could leverage other MaxSAT solvers.
>
> _’’ -I failed to find a supplementary PDF file that would  contain the proofs and/or extended discussion.’’_
>
> We are very sorry for that. We do not understand why the supplementary PDF file containing all the proofs is not in the folder. If this can be of any help for the reviewers, please note that thee proofs can be obtained via the following link: https://mega.nz/file/poNSABYL#TFdyo4RRgld-M4cI3ijh7CqnbYkBnfNd4eaQhHSdQa4
> (this link points to a repository that complies with the anonymity requirements).

---

### Official Review · Reviewer_ZH2P · 2021-07-13

**Rating:** 2
**Confidence:** 5

**Summary:**

This paper tries to do some explanations of random forests.

**Limitations And Societal Impact:**

1, This paper only considers the instance with Boolean feature vector, which is the major problem. In many machine learning problems, we usually meet the problem with real-valued feature vector, such as CV and NLP applications. The paper can not be used to address real-world machine learning applications by considering boolean feature vector.
2, Many places are unclear, such as what is "abductive explanations"? this notion appears many times, but no clear definition and explanation.  what is the "prime implicant"? still no explantion.
3, Why trade off? why there is a trade off between complexity and sparsity, why there is a sparsity? readers are confused by the paper.
4, the datasets are too tivial, can the paper try imagenet and NLP data sets?

**Main Review:**

This paper tries to do some explanations of random forests, and proposes majoritary reasons and minimal majoritary reasons. The problem and the idea is interesting. However, there are many limitations:
1, This paper only considers the instance with Boolean feature vector, which is the major problem. In many machine learning problems, we usually meet the problem with real-valued feature vector, such as CV and NLP applications. The paper can not be used to address real-world machine learning applications by considering boolean feature vector.
2, Many places are unclear, such as what is "abductive explanations"? this notion appears many times, but no clear definition and explanation.  what is the "prime implicant"? still no explantion.
3, Why trade off? why there is a trade off between complexity and sparsity, why there is a sparsity? readers are confused by the paper.
4, the datasets are too tivial, can the paper try imagenet and NLP data sets?

**Time Spent Reviewing:**

0.5

---

> ### Author Response · Authors · 2021-08-10
> **Response to the review by Reviewer ZH2P**
>
> Thanks for your comments.
>
> _’’However, there are many limitations: 1, This paper only considers the instance with  Boolean feature vector, which is the major problem. In many machine learning problems, we usually meet the problem with real-valued feature vector, such as CV  and NLP applications. The paper cannot be used to address real-world machine  learning applications by considering Boolean feature vector.’’_
>
> We are afraid that we miss your point here. We may easily consider data sets for which instances are real-valued vectors (i.e. $\mathbf x \in \mathbb R^d$). Actually, many of them have been considered in our experiments, please see Section 4. Attributes are binarized on-the-fly by the algorithm used to learn random forests (and available from Scikit-Learn library).
>
> _’’2, Many places are unclear, such as what is "abductive explanations"? this notion  appears many times, but no clear definition and explanation. what is the "prime  implicant"? still no explanation.’’_
>
> We get your point, and will clarify these notions in the revised version. In Boolean classification, we view any instance $\mathbf x \in$
> {0,1}$^n$ as a set of literals, or equivalently, a set {$x_i = v_i$}$_{i=1}^n$ of (Boolean) valued features. Thus, using the standard terminology in COLT, an _implicant_ of a Boolean function $f$ is a subset of literals $\mathbf t$ such that $f(\mathbf x) = 1$ for every instance $\mathbf x$ extending (or “covering”) $\mathbf t$. A _prime implicant_ is an implicant $\mathbf t$ of $f$ such that any proper subset of $\mathbf t$ is no longer an implicant of $f$. With these standard notions in hand, an _abductive explanation_ for the prediction $f(\mathbf x) = 1$ is any implicant $\mathbf t$ of $f$ covered by $\mathbf x$. If, in addition, $\mathbf t$ is a prime implicant of $f$, then $\mathbf t$ is called a _sufficient reason_.
>
> _’’3, Why trade off? why there is a trade off between complexity and sparsity, why there is a sparsity? readers are confused by the paper.’’_
>
> In a nutshell, the trade-off pointed out in the paper is as follows: direct reasons are  easy to compute but they are not parsimonious in general (they may contain many  irrelevant literals), majoritary reasons are almost as easy as direct reasons from the  computational point of view (one can derive a majoritary reason in polynomial time),  but they typically contain less irrelevant literals than direct reasons. Finally, sufficient  reasons are much more computationally demanding but they do not contain any  irrelevant literal. For more sparsity, minimal majoritary reasons can be considered instead of majoritary reasons, with a significant extra computational cost; similarly,  minimal sufficient reasons can be considered instead of sufficient reasons, but they  are really hard to be computed.
>
> _’’4, the datasets are too trivial, can the paper try imagenet and NLP data sets?’’_
>
> In our experiments we have considered datasets of significant sizes (for example, dorothea is based on 100 000 features and higgs-boson contains 68 636 instances). In our opinion, such data sets can hardly be considered as toy ones. Nevertheless, following for your suggestion, in the future we will also try imagenet and NLP data sets.

---

### Official Review · Reviewer_N7A7 · 2021-07-16

**Rating:** 7
**Confidence:** 3

**Summary:**

This work aims at providing a way of deriving explanations/reasons (in terms of features) from Boolean random forest (i.e., performing binary classification). They present three kinds of explanations (and the “minimal” variant for two of them): a direct reason is the most obvious one and directly comes from the interpretation of a single decision tree, a sufficient reason is one that is common to all trees, and a majority reason that is one related to the majority vote prediction and is appropriated for random forest models. Computational complexity is discussed for all approaches.

**Ethical Concerns:**

Not relevant.

**Limitations And Societal Impact:**

Limitations are well addressed this work should not have any negative social impact.

**Main Review:**

Overall comment:
The paper is well-written, and tackles a very interesting topic. I believe it provides an interesting approach to better understand a random forest model. However, I think that a few tree-specific questions are not addressed in the paper, and the work could benefit from clarifications and more links with related works (in the sense, aiming at deriving explanations/reasons for instance from tree-based models).

Comments:
- Proposed approaches is interesting and well presented. I believe the rational behind the majority reason is sound and original.
- It is now well-known that hyper-parameters of random forest (in particular the “max_features” randomization parameter controlling the feature subsampling) strongly impacts the way trees of the forest are built and how (and which) features are used. Disregarding the bagging, trees without feature subsampling focus on the most important features (in terms of impurity decrease) while trees with maximal feature subsampling (split features are randomly selected) will use more irrelevant features but also be able to discover all relevant features (thus all explanations/reasons). Similar questions have been addressed regarding the maximal depth of trees that could impact the size of explanations/reasons (i.e., the maximal number of relevant features that are simultaneously used to make a prediction). I think this work could benefit from considering the impact of such hyperparameters on the identified reasons, or at least better justify the choices made in the experiments (e.g., "why limiting the tree depth to 8?" "Why keeping default hyperparameter values?")
-  As the notion of “(irr)relevance” has been widely used in tree-based feature importance (i.e., associating each feature to an importance score) works, I think it could be useful to define the notion of relevance (or irrelevance) to clarify what is actually used here.
- I think many links could be done with tree-based (local / per sample) feature importance methods and this work surely could benefit from a short discussion as the one made for the model agnostic method LIME.
- I think it would be more convincing to also compare (or discuss) the explanations/reasons that have been found with the proposed approach(es) and those with LIME with more details than just a size and computational basis.


**Time Spent Reviewing:**

8

---

> ### Author Response · Authors · 2021-08-10
> **Response to the review by Reviewer N7A7**
>
> Thanks for your comments.
>
> _’’I think this work could benefit from considering the impact of such hyperparameters  on the identified reasons, or at least better justify the choices made in the  experiments (e.g., "why limiting the tree depth to 8?" "Why keeping default  hyperparameter values?")’’_
>
> Thanks a lot for these very interesting suggestions. Following them, in our future work, we plan to investigate the impact of hyperparameters on explanations  (measuring the numbers of explanations, the distribution of their sizes, and their  diversity in terms of features occurring in explanations).
>
> The choice of the default values for almost all hyperparameters was done to ensure a fair empirical evaluation.  The number of trees in the forest was tuned to ensure that the accuracy of the forest  is good enough (indeed, we are interested in explaining the predictions achieved by  classifiers that are sufficiently good, focusing on bad classifiers for the explanation  purpose would not be meaningful).
>
> _’’As the notion of “(irr)relevance” has been widely used in tree-based feature  importance (i.e., associating each feature to an importance score) works, I think it  could be useful to define the notion of relevance (or irrelevance) to clarify what is  actually used here.’’_
>
> You are right. When abductive explanations are concerned, ''irrelevant’’ means  ''redundant’’: a feature is irrelevant in an explanation when removing the feature from  the explanation does not question the explanatory status of the latter. We will add a sentence for clarifying it.
>
> _’’I think many links could be done with tree-based (local / per sample) feature  importance methods and this work surely could benefit from a short discussion as the  one made for the model agnostic method LIME. I think it would be more convincing to  also compare (or discuss) the explanations/reasons that have been found with the  proposed approach(es) and those with LIME with more details than just a size and  computational basis.’’_
>
> Agreed. Both types of explanations are local ones (i.e., they are instance dependent), but a fundamental difference is that the explanations we focus on are model-based, rigorous explanations, while LIME explanations are model-agnostic, heuristic explanations (see e.g. https://arxiv.org/abs/2012.11067). We plan to explain this in more details in the next version of the paper.

---

### Official Review · Reviewer_afJi · 2021-07-21

**Rating:** 9
**Confidence:** 5

**Summary:**

The paper formalizes the problem of extracting explanations from random forests, provides some clear categorizations of different types of explanations, and eventually tractable algorithms for finding not optimal but good minimal explanations.


**Ethical Concerns:**

I don't have any ethical concerns with this paper.

**Limitations And Societal Impact:**

I don't have any suggestions.

**Main Review:**

I really enjoyed reading this paper. Admittedly, the very clear formalization of the problem is probably not due to this work, and I could have known it from prior works, but I didn't. On top of that are very clear new definitions of sufficient reasons, which are expensive to find, and majoritary reasons, which are easy to find (essentially just a bottom-up greedy search). Finding minimal majoritary reasons is harder, but a polynomial solution is presented based on encoding the problem as a SAT problem.

I also liked that the paper even included a comparison to a slightly adapted version of LIME.

A minor comment: I, for one, did not know what a "prime implicant" is. It's not hard to guess, and even easier to google, but I think it wouldn't hurt to briefly mention it in the paper (I would, e.g., expect that more people in machine learning know how to convert a decision tree to DNF, which is explained in the paper, than people know what a prime implicant is).

**Time Spent Reviewing:**

1.5 hours

---

> ### Author Response · Authors · 2021-08-10
> **Response to the review by Reviewer afJi**
>
> Thanks for your comments. In the next version of our paper, we will recall what a prime implicant of a Boolean  function is and add a reference (this goes back to Quine).

---

### Decision · Program_Chairs · 2021-09-27

**Decision:**

Reject

**Comment:**

The paper concerns interpretability of the decisions made by the random forest algorithm, by considering different types of reasons that would explain why an input instance is classified as positive or negative by a Boolean random forest.

The paper received a mixed set of reviews, ranging from very positive to very negative ones.

On the positive side, some reviewers praised a clear and sound formalization of extracting the explanations from random forest through the perspective of formal logic, and generally found the proposed approach interesting.

On the negative side, however, the paper was found to have a small contribution from the formal logic point of view. Also, the presentation of the main concepts results was found to be unclear, and their discussion with the existing results lacking. Furthermore, some of the results (such as Propositions 1 and 2) felt unrelated to the rest of the paper (and only used in the proofs omitted from the main text), while some other already known.

One of the reviewers did not like the fact that the paper did not test the proposed approach for large scale data sets (there was also an issue with the applicability beyond Boolean features, but I believe it was entirely clarified in the rebuttal). Also the presentation was found unclear at places.

Overall I believe the paper would require a major revision before publishing. I would advise the authors to take into account the remarks of the reviewers, especially reviewer 4ukE, while preparing such revision.